# Fundamental and Advanced Therapies, Vaccine Development against SARS-CoV-2

**DOI:** 10.3390/pathogens10060636

**Published:** 2021-05-21

**Authors:** Nikola Hudakova, Simona Hricikova, Amod Kulkarni, Mangesh Bhide, Eva Kontsekova, Dasa Cizkova

**Affiliations:** 1Centre for Experimental and Clinical Regenerative Medicine, The University of Veterinary Medicine and Pharmacy, 04181 Kosice, Slovakia; hudakova.nikolka@gmail.com; 2Laboratory of Biomedical Microbiology and Immunology, The University of Veterinary Medicine and Pharmacy, 04181 Kosice, Slovakia; simon.hricik@gmail.com (S.H.); kulkarni.adm@gmail.com (A.K.); bhidemangesh@gmail.com (M.B.); 3Institute of Neuroimmunology, Slovak Academy of Sciences, 84505 Bratislava, Slovakia; eva.kontsekova@savba.sk

**Keywords:** SARS-CoV-2, COVID-19, cytokine storm, therapy, antibodies, vaccines

## Abstract

Coronavirus disease (COVID-19) caused by the SARS-CoV-2 virus has been affecting the world since the end of 2019. The severity of the disease can range from an asymptomatic or mild course to acute respiratory distress syndrome (ARDS) with respiratory failure, which may lead to death. Since the outbreak of the pandemic, scientists around the world have been studying the genome and molecular mechanisms of SARS-CoV-2 infection to develop effective therapies and prevention. In this review, we summarize the progressive development of various treatments and vaccines as they have emerged, a year after the outbreak of the pandemic. Initially for COVID-19, patients were recommended drugs with presumed antiviral, anti-inflammatory, and antimicrobial effects that were previously used to treat other diseases. Thereafter, therapeutic interventions were supplemented with promising approaches based on antibodies, peptides, and stem cells. However, licensed COVID-19 vaccines remain the most effective weapon in combating the pandemic. While there is an enormous effort to enhance the vaccination rate to increase the entire population immunity, the production and delivery of vaccines is becoming limited in several countries. In this regard, there are new challenges needing to be addressed by combining non-pharmacological intervention with effective therapies until vaccination is accessible to all.

## 1. COVID-19 Outbreak

Since the first cases reported from Wuhan (Hubei Province of China) at the end of 2019, there has been an expansion of the severe acute respiratory syndrome coronavirus-2 (SARS-CoV-2), previously named as novel coronavirus or 2019-nCoV [1], in all continents, including Antarctica [2]. At the beginning of the outbreak, an epidemiological investigation in Wuhan identified an initial association with a seafood market selling live animals [3]. Nowadays, a new data describing molecular and serological evidence of SARS-CoV-2 related coronaviruses in bats occurring in China denote a high possibility of bat-to-human transmission [4].

However, the main means of SARS-CoV-2 transmission overall is from person to person by inhalation of smaller-than-droplet particles (airborne route) [5,6]. From the beginning, non-pharmaceutical recommendations, such as strict hand hygiene, wearing face mask, safe social distancing, and compliance to quarantine, were shown to be effective in controlling the spreading of infection. The virus has been detected also in non-respiratory samples (e.g., blood, stool); however, a role of these biological materials in spreading is unclear [7,8,9]. There were also reports on perinatal transmission route, but whether the transmission has been transuterine, transplacental, or environmental is not determined yet [10,11].

People in their 60s or 70s are generally more susceptible to SARS-CoV-2. Thus, the severity of the disease is positively correlated with age and underlying diseases (hypertension, uncomplicated diabetes, cardiovascular disease, chronic respiratory disease, immune compromised status, cancer, obesity, etc.) [3,12]. The number of children infected by SARS-CoV-2 increased gradually with the rising spread of the epidemic. However, SARS-CoV-2 (like SARS and MERS) was detected in pediatric patients less frequently with milder symptoms and with a better overall outcome than in adults [3].

A broad spectrum of SARS-CoV-2 clinical manifestations in infected patients ranged from mild symptoms that were nonspecific to severe pneumonia with organ function damage [13]. The accompanying symptoms can be grouped into three clusters. The most common respiratory symptom cluster (cough, production of sputum, febrility, etc.), a musculoskeletal cluster (muscle pain, joint pain, headache, and exhaustion), and gastrointestinal (enteric) cluster (vomiting, diarrhea, and abdominal pain) [14]. A pooled analysis of five studies among 817 patients showed that gustatory malfunction (altered taste sensation) was found among 49.8% of COVID-19 patients [15]. Another study has confirmed that anosmia (impaired olfaction) in patients suffering from COVID-19 varied from 33.9 to 68% with female dominance [16,17].

## 2. The Genome and Structure of SARS-CoV-2

Coronaviruses are single-stranded unsegmented positive-sense RNA viruses with a dimension of 80–120 nm. There are four types of coronaviruses, namely, α-coronavirus, β-coronavirus, δ-coronavirus, and γ-coronavirus [18,19], in which the genome varies from 26 to 32 kilobases. They belong to the order Nidovirales, the family Coronaviridae, and subfamily Coronavirinae [18]. SARS-CoV-2 belongs to the genus Betacoronavirus [20].

Coronavirus Research Group of the International Committee on Taxonomy of Viruses (ICTV) has determined that a novel coronavirus is affiliated with the SARS virus (SARS-CoV) [21]. Phylogenetic analysis of full-length genome sequences obtained from infected patients showed 79% similarity between SARS-CoV-2 and SARS-CoV [22,23]. As both SARS-CoV and SARS-CoV-2 belong to the category called severe acute respiratory syndrome-related coronavirus, the ICTV assigned the name of this coronavirus as severe acute respiratory syndrome coronavirus 2 (SARS-CoV-2) [21]. High genome-wide sequence homology (88–89%) is also found between SARS-CoV-2 and two bat-derived SARS-like coronaviruses, namely, bat-SL-CoVZC45 and bat-SL-CoVZXC21. The sequence homology between SARS-CoV-2 and Middle East respiratory syndrome coronavirus (MERS-CoV) accounts for only 50% [23]. SARS-CoV-2 became the seventh member of the coronavirus family to infect humans [22]. The other coronaviruses are human coronavirus 229E, NL63, OC43, HKUl (HCoV-229E, HCoV-NL63, HCoV-OC43, HCoV-HKU1, respectively), SARS-CoV, and MERS-CoV [24]. Variable numbers of open reading frames (ORFs) are found in the coronavirus genome [25]. The SARS-CoV-2 genome was reported to possess 14 ORFs encoding 27 proteins, among which four encode major structural proteins localizing on the surface of SARS-CoV-2, namely, spike surface glycoprotein (S) and matrix protein (M), small envelope protein (E), and nucleocapsid protein (N). These four ORFs are located at the 3′-terminus of the genome [26] (Figure 1A).

The S glycoprotein is required for binding to receptors on the host cell and plays an essential role in determining host tropism and transmission capacity, mediating receptor binding and membrane fusion [27]. In general, spike protein consists of two functional subunits: S1 and S2 domains. While S1 domain mediates receptor binding, S2 domain is responsible for cell membrane fusion [28]. Several analyses have shown that SARS-CoV-2 uses cellular angiotensin-converting enzyme 2 (ACE-2) as its receptor for binding to host cells [29,30,31,32,33]. The cleavage site present between S1 and S2 on protein S is proteolytically cleaved by cellular cathepsin L and the transmembrane protease serine 2 (TMPRSMP2) [34]. While TMPRSS2 exposes the surface of the plasma membrane of the host cell, cathepsin L activates S protein in endosomes. Cathepsin L can also compensate the entry into cells devoid of TMPRSS2 [31]. The cleavage site S1/S2 uniquely disposes of the insertion RRAR, located between residues 682 and 685 [30,32,33]. Due to this insertion, S1/S2 can be also pre-cleaved by furin, which reduces the dependence of SARS-CoV-2 on TMPRSS2 and cathepsin L on host cells [30,32]. Considering the fact that S protein can be uniquely pre-cleaved by furin, which is found in almost all tissues and organs, SARS-CoV-2 is able to induce systemic infection. It can therefore be much more infectious compared to other types of SARS-like coronaviruses [33] (Figure 1C).

Although possible mutations were not initially considered, nowadays, much attention has been paid to the health risks of newly discovered variants of the SARS-CoV-2 virus [35]. Both SARS-CoV-2 variants, 501Y.V1 (British B.1.1.7) identified across the UK, except Northern Ireland, and 501Y.V2 (B.1.351) spreading in South Africa, have a mutation (N501Y) in the receptor-binding domain (RBD) of the spike protein, which makes them more transmittable (40–70%) in comparison with other variants [36]. Moreover, the South African and Brazilian (P.1) variants have shown a similar capability of adapting to evade immunity as well as antibody escape [37]. It is still not clear as to whether immunity provided by T cells may protect the organism against these mutations. Therefore, the effect of these mutations on transmission, severity of the COVID-19 infection, and vaccination strategies is currently the subject of numerous scientific studies [35].

It is necessary to note that while the development of coronavirus vaccines and global vaccination is currently underway, the immediate effective treatment that would prevent a serious course of COVID-19 in patients still remains as the main issue. Therefore, various fundamental and experimental therapies involving drugs, antibodies, peptides, or even stem cells with different mechanisms of action have been tested. Since the beginning of the pandemic, safe treatment protocols are gradually being introduced for those confirming efficacy in inhibiting SARS-CoV-2 infection.

## 3. Treatment Strategies for SARS-CoV-2 Infection

### 3.1. Fundamental Therapies

In early 2020, no officially approved treatment for COVID-19 was available, and thus the obvious option at hand was to test already approved drugs with antiviral, anti-inflammatory, and antimicrobial properties. Although the safety and toxicity of these drugs were well known, the efficacy and doses required against SARS-CoV-2 infection were challenging to determine. Furthermore, their mechanism of action against the virus was often speculative because of the lack of substantial clinical studies [40]. In agreement with the advice from the New and Emerging Respiratory Virus Threats Advisory Group (NERVTAG, UK), six drugs and two biological therapies (immunoglobulins and synthetic neutralizing antibodies) entered into a randomized trial investigating the efficacy of these pharmaceuticals to prevent death in COVID-19 patients (RECOVERY trial) [41]. We mention only some that have been tested to combat SARS-CoV-2 infection (Figure 2).

#### 3.1.1. Antimalarial Drugs

Chloroquine (CQ) and hydroxychloroquine (HCQ—hydroxy-analogue of CQ) are older antimalarial drugs. These molecules have long been used worldwide as drugs of choice for the prophylaxis and treatment of malaria with known safety and efficacy [42]. Increase in endosomal pH could inhibit viral/endosomal membrane fusion, which is necessary for the release of the viral genome into the host cell (Figure 2). In addition, other mechanisms of action also include impairment in endosome mediated virus entry; phagolysosomal fusion; replication of viral nucleic acid; glycosylation of newly synthesized viral proteins; assembly of virus; release of progeny through exocytosis; inhibition of the host cytokine storm; and autophagy modulators that interact with coronaviruses, retroviruses, and flaviviruses [43,44]. CQ and HCQ significantly reduced SARS-CoV and SARS-CoV-2 infection in vitro [44], and these promising initial results led to the emergency usage authorization of CQ/HCQ from the FDA in March 2020 [45,46]. However, hydroxychloroquine failed to reduce the mortality in patients hospitalized with COVID-19 [47], and chloroquine derivatives have immunosuppressive and adverse cardiac effects [48]. This led to a decision of the World Health Organization (WHO) and the Food and Drug Administration (FDA) for not supporting the use of chloroquine and hydroxychloroquine for COVID-19 patients [40].

#### 3.1.2. Remdesivir

Remdesivir (GS-5734) is a new broad-spectrum antiviral drug of small molecular weight, one that is an analogue of adenine nucleotide [49,50]. GS-5734 is a prodrug that is metabolized into GS-441524, an active form of remdesivir interfering in the activity of viral RNA-dependent RNA polymerase (RdRp) [50,51]. As proven earlier, remdesivir is effective against paramyxovirus, filovirus, and coronavirus infections [50]. The antiviral potency of remdesivir is directed towards the suppression of RdRp activity, leading to cessation of viral RNA synthesis (Figure 2) [52].

RdRp is required for the genome replication of SARS-Cov-2 in host cells [49]. The molecular docking study confirmed that remdesivir is able to bind the model of SARS-CoV-2′s RdRp very well and contradict the RdRp function [53]. Thus, inhibitors of the RdRp such as remdesivir could be used for this purpose [29].

Remdesivir has shown antiviral activity against SARS-CoV and MERS-CoV in human airway epithelial cell lines and was able to reduce the load of SARS-CoV in a mouse model [54]. Successful inhibition of COVID-19 infection was detected in Vero E6 cells and human hepatocellular carcinoma cells (HuH-7), which are highly sensitive to SARS-CoV-2 because of the prominent number of ACE-2 receptors expressed on its surface [51]. Due to its safety and efficacy in resisting corona viruses, remdesivir was suggested as a suitable therapeutic for the treatment of SARS-CoV-2 [55]

In a clinical study, patients suffering from COVID-19 receiving remdesivir have shown clinical improvement in terms of reduced oxygen support [56]. However, the effect of the use of remdesivir on baseline viral load and viral suppression has not been determined [56].

Two large studies, ACTT-1 and SOLIDARITY, were published on the use of remdesivir in patients with COVID 19 but showed different results. In the case of ACTT-1, beneficial effects were observed in terms of shortening the time of clinical improvement and reducing mortality in patients who received supplemental oxygen, but not in those with supportive ventilation. On the other hand, in the case of solidarity, there was no reduction in mortality with remdesivir [57,58]. Although the results of the two studies are different, it is impossible to compare them objectively because they had different study designs and focused on different parameters such as clinical improvement, mortality, glucocorticoid administration, and different geographical distribution of patients.

#### 3.1.3. Glucocorticoids

Glucocorticoids, such as dexamethasone, prednisone, methylprednisolone, and hydrocortisone, may regulate inflammation-mediated lung injury and therefore reduce the progress to respiratory failure, and subsequently death. The patients suffering from severe COVID-19 develop a hyper-inflammatory state also named cytokine storm. The nsp5-3C-like proteinase on SARS-CoV-2 suppresses the transport of HDAC2 (histone deacetylase 2) into the nucleus, and therefore impairs the way in which it mediates cytokine responses and inflammation. Activation of HDAC2 by dexamethasone may directly oppose the SARS-CoV-2 activity [59,60,61,62].

Preliminary results from the RECOVERY trial showed that the use of dexamethasone for up to 10 days was followed by lower 28-day mortality among hospitalized COVID-19 patients than in random group of individuals receiving the usual care of invasive mechanical ventilation, or those receiving oxygen without invasive mechanical ventilation.

On the contrary, among patients not receiving respiratory support, there was no evidence of benefit from dexamethasone use [63].

United Kingdom chief medical officers, National Institutes of Health (NIH) in the USA, and the World Health Organization (WHO) have improved COVID-19 treatment guidelines and recommended the use of dexamethasone in hospitalized patients with COVID-19 and respiratory failure requiring therapy with supplemental oxygen or mechanical ventilation [63].

#### 3.1.4. Ivermectin

Ivermectin, a traditional antiparasitic drug, is well known to inhibit importin α/β1-mediated nuclear import of viral proteins leading to inhibition of dengue and HIV1 virus replication in vitro [64]. Likewise, ivermectin-based inhibition of other flaviviruses such as West Nile virus, tick-borne encephalitis virus, and yellow fever virus replication in Vero cells has been demonstrated [65]. Of late, this FDA-approved anti-parasitic drug has also been tested in phase III clinical trial of patients suffering from Dengue virus infection in Thailand (https://clinicaltrials.gov/ct2/show/NCT02045069 accessed on 4 March 2021). Several single-stranded RNA viruses, including two coronaviruses, are known to localize their nucleocapsid protein in the nucleous/nucleolus of infected host cells to mediate damping of the cell cycle and antagonize the interferon signaling pathway [66]. Currently, it is unclear if nucleocapsid proteins of SARS-CoV-2 enter the nucleus, but the virus is known to disrupt interferon signaling by avoiding the translocation of STAT-1 into the nucleus [67]. Therefore, members of the importin super family will have a pivotal role in transferring macromolecules across the nuclear envelop [68]. On the basis of the aforestated factors, researchers evaluated ivermectin’s antiviral activity for SARS-CoV-2 on Vero/hSLAM cells with a single dose of 5 µM/mL. As a result, reduced viral replication within 24–48 h was noticed [69]. The in vitro study of Caly et al. [69] used a very high dose of ivermectin, which was safe. The absence of any other effective treatment for SARS-CoV-2, resulted in the initiation of ivermectin clinical trials on COVID-19 patients in several countries of Latin America [70].

As of now, there have been 56 clinical trials in which ivermectin was used in hospitalized patients along with the standard treatment or in combination with antibiotics (azithromycin, doxycycline)/steroids/hydroxychloroquine/alternative therapy (https://www.covid-trials.org/ accessed on 4 March 2021). Authors of a clinical trial in Bangladesh have claimed that ivermectin can be useful to treat patients with mild to moderate COVID-19 disease [71]. On the other hand, ivermectin was shown to lower the mortality in patients with COVID-19 that were having severe pulmonary involvement [72]. In particular, the clinical trial of Chaccour et al. [73] showed a decrease in anosmia/hyposmia, reduced viral loads, and lower IgG titers (although not statistically significant) in patients treated with ivermectin, proposing the mechanism of action of ivermectin against COVID-19. The authors predicted that ivermectin can cause positive allosteric modulation of the nicotinic acetylcholine receptor. As a result, the expression of ACE-2 receptor would decrease, leading to reduced viral entry into the cells of the respiratory epithelium and olfactory bulb [73]. Moreover, ivermectin is known to downregulate the expression of pro-inflammatory genes, IL-8, TNF-α, and cathelicidin LL-37 [74]. Similar downregulation of pro-inflammatory pathways in the olfactory epithelium could have reversed COVID-19-induced anosmia [73]. Regardless of the positive aspects of ivermectin and its approval in Slovakia and a few more European countries, the panel of COVID-19 treatment guidelines, National Institute of Health, USA, has declared that the outcome of present clinical trials are not adequate enough to recommend ivermectin for the treatment of COVID-19. Therefore, further well-designed and well-conducted clinical trials are warranted.

#### 3.1.5. Passive Immunity Therapy

Since the Spanish flu era, convalescent blood products taken from a surviving patient have been transfused to treat influenza A (H5N1), measles, hepatitis A, hepatitis B, chickenpox, and rabies virus infections [75,76,77,78,79]. Passive immunization and immunotherapy refer to the transfer of the acellular portion of human blood containing antibodies to a vulnerable individual for the prevention/treatment of disease [80]. Convalescent plasma contains antibodies that act through different mechanisms. They can minimize the virulence of virus by direct binding, or they can promote distinct host immune pathways—complement activation, antibody-dependent cytotoxicity, or phagocytosis [81]. Non-neutralizing antibodies may contribute to prophylaxis and speed up recovery with their ability to bind to the pathogen but not interfere with its replication in in vitro systems [82]. This was confirmed in a pilot study reporting improvement in clinical symptoms and laboratory parameters of COVID-19 patients after they received convalescent plasma. Some of the noticeable changes showed an increase in serum neutralizing antibody titers, alleviation of inflammation and/or over-activated immune response, and absence of viral RNA in tested blood samples [83]. A randomized controlled clinical trial involving intravenous transfusion of immunoglobulin (IVIg) (https://clinicaltrials.gov/ct2/show/NCT04261426 accessed on 30 January 2021) indicated that passive immunization could interrupt the storm of inflammatory factors and promote the activity of T lymphocytes and B lymphocytes in peripheral blood [84].

Despite the successful results, convalescent plasma has several limitations including the requirement for matching of blood type and batch-to-batch variability, as well as screening for various pathogens (HIV, parasites, hepatitis).

A retrospective study of 3082 patients indicated that transfusion of plasma containing higher levels of anti-SARS-CoV-2 IgG antibodies was associated with lower risk of death compared with transfusion of plasma containing lower levels of antibodies [85].

In contrast, the PLACID trial focusing on the effectiveness of using convalescent plasma for treating COVID-19 did not show any positive effect on reducing all-cause mortality [86]. These data correlate with a recent meta-analysis that concluded that there was no significant association with a decrease in all-cause mortality in patients receiving plasma compared with placebo or with standard care [87].

Monoclonal antibodies appear to be a suitable replacement. A number of modern techniques (e.g., animal immunization, phage/yeast display, antigen-specific B cell sorting) can be used to obtain antiviral monoclonal antibodies or their derivatives relatively quickly [81]. Suitable candidates could be combinations, e.g., bamlanivimab/etesevimab and casirivimab/imdevimab, which have authorization for administration in the treatment of mild to moderate COVID-19 infections in pediatric patients (at least 12 years old), in adults with laboratory-confirmed SARS-CoV-2 infection, and also in individuals at high risk of admission to hospital or progressing to severe COVID-19 and ARDS [88].

##### Antibody-Dependent Enhancement of Diseases (ADE)

Transfusion of convalescent plasma or antibody-based passive immunization seems to have beneficial effects in prophylaxis at the onset of symptoms to neutralize the virus [83,89].

Nevertheless, there are pessimistic concerns over the usage of antibody therapies that can theoretically amplify the infection or trigger harmful immunopathology. This phenomenon is referred to as antibody-dependent enhancement of diseases (ADE), and the same has been recognized for S protein-based vaccine candidates for SARS and MERS [90,91,92]. For instance, non-human primates treated with SARS-CoV S protein–IgG during SARS-CoV infection developed acute lung injury in the early phase of the disease and lost wound-healing response and production of TGF-β [90]. Likewise, hypersensitive lung pathology, namely, increased infiltration of eosinophils releasing IL-5 and IL-13 cytokines, was noticed in murine models immunized with inactivated MERS-CoV vaccine [92]. Evidenced of ADE has also been noticed for mosquito-borne flaviviruses, i.e., Zika virus and dengue infection [93,94]. ADE is prominently observed when the antibodies generated or received by a recipient are of sub-/non-neutralizing types. The virus-sub-neutralizing antibody immunocomplex could be efficiently phagocytosed by myeloid cells through the Fcγ receptor, leading to enhanced virus infectivity. On the other hand, the internalized immunocomplexes can modulate the excessive release of inflammatory and vasoactive mediators, resulting in disease severity [95]. Although there are no studies at the moment validating the phenomenon of ADE for SARS-CoV-2 in animal models, an in vitro study on lymphoma cells (Raji cells) has shown enhanced COVID-19 infection in the presence of patient plasma mediated by IgG antibodies through FcγRII receptor [96]. In this context, it becomes obligatory not just to evaluate the efficacy and safety pharmacology of newly developed anti-SARS-CoV-2 preventive measures such as vaccines, but also to assess the potential risk of ADE.

#### 3.1.6. Supportive Therapies

COVID-19 is a multi-system disease affecting various organs, and due to this fact, it has become evident that supportive therapy is crucial in the management of this disease. A striking feature of COVID-19 is the high incidence of thromboembolism. Although the exact mechanism is not clear, it has been hypothesized that an amplified inflammatory response to SARS-CoV-2 virus could lead to endothelial dysfunction and venous and arterial thromboembolic events [98]. Low-molecular weight heparins (LMWHs) appear to be suitable candidates as anticoagulants for all hospitalized adults with COVID-19 as they inhibit activated factor Xa via a complex with antithrombin. Several ongoing phase III randomized clinical studies are investigating whether high-dose versus low-dose anticoagulants more effectively reduce the risk of thrombosis (NCT04345848), as well as evaluating the efficacy of therapeutic-dose LMWH versus prophylactic and intermediate dose of LMWH or unfractionated heparin for prevention of venous and arterial thromboembolic events and mortality (Hep-COVID) [99,100].

Another serious feature of COVID-19 infection is hypoxia. Inhaled pulmonary vasodilators tend to improve oxygenation as they reduce pulmonary vascular resistance and lower the vascular pressure in the lungs. A retrospective study involving seven patients showed that the inhalation of nitric oxide significantly improved oxygenation in patients with ARDS [101].

The effectiveness of bronchodilators in the management of COVID-19 infection is still unclear. However, the disease is mainly transmitted by the airborne route, and therefore the use of nebulizers that involve a risk of accelerating transmission is debatable. Patients with chronic obstructive pulmonary disease (COPD) are advised to preferably use metered dose inhalers with spacer devices [99].

The majority of COVID-19 patients are asymptomatic or have mild respiratory symptoms, but there is a serious concern about their developing acute respiratory distress syndrome (ARDS), which requires hospitalization and intensive care. For these patients, supplemental oxygenotherapy in the form of high nasal flow ventilation during mild or moderate ARDS and mechanical ventilation for serious ARDS cases is crucial. Extracorporeal membrane oxygenation (ECMO) treatment has shown ability to increase critically ill patients’ survival rate where no other treatment strategy could have been used [99].

### 3.2. Advanced and Experimental Therapies

#### 3.2.1. Peptides

In terms of the search for an effective antiviral therapy against COVID-19, antiviral peptides, otherwise known as antimicrobial peptides, could assume the role as one of the potential classes of new anti-SARS-CoV-2 therapeutics. Antiviral peptides, mainly composed of short amino acid sequences, can specifically target various viral components to achieve convincing antiviral effects [102]. According to the latest studies, antiviral peptides with anti-SARS-CoV-2 activities are mucroporin-M1, HR2P-M2, EK1/EK1C4, P9 peptide, RTD-1, and HD5. Their antiviral actions are rather diverse, such as disrupting the viral envelope (M1), inhibiting viral fusion mechanism (HR2P-M2), blocking viral HR1 domain (EK1/EK1C4), preventing the release of viral RNA (P9), and also conferring immune activation in the host (RTD-1) and preventing a virus from attachment to ACE-2 receptor (HD5) [31,102,103]. There is also some evidence highlighting the role of the recently identified S protein furin-like cleavage site and viral entry pathway mediated by CD147 in SARS-CoV-2 pathogenicity [104,105]. These important viral segments could be valuable targets for the development of new antiviral agents. A combined therapy using a mix of antiviral peptides or other classes of antiviral agents could be used as supplemental medications [106]. Overall, due to their relatively simple primary structure, antiviral peptides are functionally versatile and are possible molecular templates for the generation of original therapeutic candidates against emerging global threats such as COVID-19 [102].

#### 3.2.2. Angiotensin-Converting Enzyme Inhibitors/Angiotensin Receptor Blockers

Clinical studies have portrayed hypertension as a risk factor in patients suffering from SARS compared to normotensions. In the treatment of patients with COVID-19 with hypertension, the use of angiotensin-converting enzyme inhibitors or angiotensin receptor blockers (ACEIs or ARBs) has been controversial [107,108]. These therapeutic interventions involve ACE-2, the cellular receptor targeted by SARS-COV-2 for its entry and propagation in host cells [109]. ACE-1 and ACE-2 are angiotensin-converting peptides. ACE-1 generates angiotensin (Ang) II from angiotensin I, causing vasoconstriction, inflammation, and increased vascular permeability, and, therefore, speeds up the development of acute respiratory disease syndrome (ARDS) in patients infected with SARS-CoV-2. The mechanism of ACE-1 signaling is mediated by G-protein-coupled receptors, namely, AT1 and AT2. The function of AT1 (generated by ACE-1) is crucial for mediating the actions of Ang II and, therefore, opposing the actions of ACE-2-derived peptides. ACE-2 is a zinc metalloprotease, responsible for Ang II conversion to Ang (1–7) [110]. The SARS-CoV-2 virus binds to ACE-2, which leads to a reduced conversion of Ang II to ACE-2-derived peptides and, therefore, a decrease in Ang (1–7) and their actions opposing the effects of Ang II. The disbalance between ACE-1 and ACE-2 actions results in the development of more severe pathological processes. After spike protein binds with ACE-2, the amount of cell surface-expressed ACE-2 is reduced. The downregulation of ACE-2 receptor will lead to worsening of lung failure [111,112].

Angiotensin-converting enzyme inhibitors (ACEIs) and angiotensin receptor blockers (ARBs) are drugs interfering with the renin angiotensin aldosterone system (RAAS) that have long been used to treat cardiovascular and renal diseases, as well as hypertension. The mechanism of action of these drugs (lisinopril, captopril, ramipril, perindopril, trandolapril, enalapril, losartan, valsartan, etc.) on COVID-19 is slightly different, as they do not affect the ACE-2 receptor directly. ACEIs increase the level of intestinal mRNA of ACE-2, but this was not observed with ARBs [113]. While the expression of attached ACE-2 is in direct correlation with the severity of COVID-19, it is not the case with the free circulating form. The free ACE-2 form could inactivate SARS-CoV-2 by preventing its entry to the pulmonary endothelium, and therefore soluble human recombinant ACE-2 could act as a protective molecule in the development of severe ARDS with subsequent clinical manifestations and death. Therefore, the lung pathogenicity of COVID-19 may be predicted by a ratio between attached ACE-2 expression/availability and circulating ACE-2 [23].

Studies on rats have suggested that the above-mentioned ACEIs/ARBs could increase the expression of ACE-2 receptor [114,115]. On the other hand, research on mice and humans showed just the contrary [116,117]. Guo et al. disproved the hypothesis that there was a higher risk of severe COVID-19 infection in patients using ACEIs/ARBs [118,119]. Recent studies also indicated that ACEIs/ARBs may be beneficial in the prevention of viral infection. Supposedly, the inhibition of ACE by ACE inhibitors may result in negative feedback (given the lack of Ang II and ACE-2 upregulation, as well as the decrease in inflammation) [120]. Recent studies also speculate that ACEIs/ARBs could prevent viral entry of SARS-CoV-2 by stabilizing ACE-2–AT1R interaction and preventing viral protein–ACE-2 interaction or internalization. The interaction of viral protein with ACE-2 decreases in the proximity of stabilized complexes ACE-2–AT1R [108].

#### 3.2.3. Mesenchymal Stromal Cell-Based Therapy for COVID-19

Mesenchymal stem (stromal) cells (MSC) that were included in pre-clinical as well as clinical studies showed beneficial effects in the treatment of respiratory tract diseases from both infectious and noninfectious causes [121,122,123]. These encouraging data led to the launch of many MSC-based treatments for COVID-19 patients that are either completed or are currently underway (reviewed by Liu et al. (2020)) [124]. To date, there are more than 22 clinical studies based on the use of MSC derived from umbilical cord blood, Wharton’s jelly, bone marrow, dental pulp, or other human origins. In addition, two clinical trials based on MSC-derived exosomes are also registered in this regard [125] (https://clinicaltrials.gov/ct2/show/NCT04276987 accessed on 23 January 2021, https://clinicaltrials.gov/ct2/show/NCT04491240 accessed on 23 January 2021, http://www.chictr.org.cn/ accessed on 23 January 2021).

According to published clinical data, the trials conducted thus far have reported that cell-based MSC treatment with human umbilical cord (UC) MSC and bone marrow MSC are safe and effective, especially for the treatment of critically ill COVID-19 patients [126,127]. Systemic delivery of MSC improved the vital functions of patients and reduced viral titers just a few days after initiating the treatment. Furthermore, the hematological and immune profile is normalized by the decrease of white blood cells (WBC) and neutrophils, as well as the increase of the number of peripheral lymphocytes. The C-reactive protein (CRP), pro-inflammatory cytokine TNF-α, and IL-6 decreased, while the anti-inflammatory protein interleukin-10 was elevated [127]. It is believed that most of the beneficial effects of MSC-based therapy are attributed to their immunomodulatory, regenerative, antimicrobial, and antiviral properties, ultimately leading to lung regeneration [125].

Majority of COVID-19 clinical trials use intravenous delivery of MSC, which are temporary trapped inside lung tissue, being the prime site of SARS-CoV-2 pathology. In critically ill patients, severe pneumonia is associated with excessive and uncontrolled inflammatory responses that trigger a cytokine storm pathology [128,129]. Overproduction of immune cells (T-helper (Th)17, CD8^+^ T cells, dendritic cells), pro-inflammatory cytokines (IL-6, IL-1β, TNF-α), and chemokines (CXCL10 and CCL2) may lead to irreversible damage of epithelial and endothelial cells, vascular leakage, and acute respiratory distress syndrome (ARDS) [130].

In this scenario, the beneficial action of MSC-based therapy is attributed to their capability to interact with released pro-inflammatory cytokines through their receptors and stimulate the biosynthesis of complex immunomodulatory molecules such as IL-10, transforming growth factor-*β* (TGF-β), TNF-*α*-stimulated gene/protein 6 (TSG-6), superoxide dismutase (SOD), cyclooxygenase-2 (COX-2), prostaglandin-E2 (PGE2), and indoleamine 2,3 dioxygenase (IDO), which, by acting via different pathways, redirect immune cells toward an anti-inflammatory phenotype [131]. In addition, MSC regulates phagocytosis and tissue regeneration by macrophage polarization from an inflammatory M1 phenotype into an anti-inflammatory M2 phenotype [132]. All those bioactive molecules together frame an anti-inflammatory environment with a predominance of Treg cells and reduced cytokine storm profile [129,133]. Thereby, MSC reveal a potential to control exacerbated inflammation, not only in affected lung as the prime site of injury, but also in the heart, kidneys, or intestinal microenvironment [134,135].

Furthermore, protection and regeneration of alveolar epithelial cells may be promoted by the MSC-released paracrine molecules, particularly those with proangiogenic and antiapoptotic efficacy such as angiopoietin 1 (ANGPT1), epidermal growth factor (EGF), vascular endothelial growth factor (VEGF), keratinocyte growth factor (KGF), and hepatocyte growth factor (HGF) [136]. Other MSC-derived paracrine mediators are contributing to extracellular matrix (ECM) remodeling and to tissue healing with decreased scarring processes [137]. To date, an increasing number of studies suggest that many of these paracrine effects are also mediated via small extracellular vesicles (EVs) recognized as exosomes and microvesicles included in the MSC secretome [138]. MSC-derived EVs (MSC-EVs) are plasma membrane structures that carry lipids, proteins/peptides, DNA, mRNA, and non-coding microRNAs [139]. In particular, miRNAs such as Let-7, miR-34a, miR-2b/c, and miR-146 are implicated in downregulation of IL-6, reduction of complement induced cytolysis, and regulation of NF-kB, and thus are taken as a whole, exerting anti-inflammatory and cytoprotective properties [139,140,141].

Interestingly, mitochondrial transfer from MSC to immune cells and respiratory epithelial cells has also been described. This unique intercellular transmission mechanism led to downregulation of inflammation and recovery of aerobic respiration in lungs Court et al., 2020, Han et al., 2020.

MSC may affect secondary bacterial infection manifested during or after viral infection through the secretion of antimicrobial factors, such as peptide LL-37 and lipocalin-2. Both promote migration and phagocytosis of macrophages, leading to pulmonary bacterial clearance [142].

Some studies point to possible antiviral mechanisms of MSC. Especially, undifferentiated progeny of MSC express constitutively elevated levels of specific interferon (IFN)-stimulated genes (ISG) including interferon-induced transmembrane family (IFITM) proteins (IFI6, ISG15, SAT1, PMAIP1, p21/CDKN1A). These proteins are capable of preventing viruses from crossing the lipid bilayer of the host cell and accessing the cytoplasm as well as blocking mRNA transcription, nuclear transport, amplification, and virus assembly and release [143,144]. In addition, pro-inflammatory cytokines, including IFN-γ, may further enhance level of antiviral proteins and induce innate defense that could lead to therapeutic benefits in COVID-19 patients. Thus, MSC interferon regulatory mechanisms may include both intrinsic (constitutive antiviral proteins) and inducible (secondary response to IFN) antiviral defense. On the other hand, it is necessary to mention that there are also studies showing that bone marrow-derived (BM) MSC can support replication of both avian H1N1 and H9N5 influenza strains; therefore, precious antiviral effects still remain to be determined [145].

In summary, due to the known and proven immunomodulatory effect of MSC, the therapy of COVID-19 patients should aim for very severe cases in which an uncontrolled immune response accompanied by cytokine storm, critical ARDS, and systemic organ pathology is developed [143,146]. Thus, it should be contraindicated at the beginning of infection when physiological inflammation is fighting against the virus [147]. Furthermore, with regards to MSC-based therapy, several important challenges need to be addressed, principally, the selection of high quality MSC, the effective route of MSC delivery, appropriate dosing and timing, and the following of ethical and moral guidelines applicable for cell-based clinical trials [148,149]. Therefore, to achieve successful MSC-cell based therapies with effective and significant results for COVID-19 patients, one must prepare well-designed, randomized, placebo-controlled, large patient cohorts and controlled clinical trials [150].

## 4. Prevention

### Vaccine Development

As vaccines are considered the most promising way to eradicate the SARS-CoV-2 virus, several teams are intensively working on vaccine development [151]. Vaccines are being developed with different technologies, some well-known and others completely new for human vaccines, such as peptide and nucleic acid technologies.

Currently, there are two messenger RNA (mRNA) vaccines and two vector vaccines to prevent COVID-19, all authorized by the European Medicines Agency (EMA). The first mRNA vaccine, Comirnaty (BNT162b2), developed by BioNTech and Pfizer, was authorized by December 2020 [152]. In January 2021, the EMA approved the COVID-19 vaccine Moderna (mRNA-1273), which was developed by the National Institute of Allergy and Infectious Diseases (NIAID) in collaboration with Moderna Biotech Spain, S.L. [153]. Both contain lipid nanoparticle (LNPs)-encapsulated mRNA, which encodes the spike protein of SARS-CoV-2 [154]. This technique does not include parts of the virus, only the sequence of spike protein encoded to mRNA. For successful delivery, novel lipid nanoparticles are used to protect the protein sequence. After intramuscular application, the LNPs are removed by myocytes and the mRNA is released and translated to endogenously synthesized spike protein. The mRNA is very sensitive, and therefore is broken down shortly after vaccination. These vaccines activate T cells cytotoxicity as well as B cells response, which ultimately causes strong cellular immunity [154,155,156,157].

The technology of adenovirus-based vectors is an already proven method of vaccine preparation, in a relatively short time, through modification of an adenovirus vector carrier through the insertion of a “gene of interest” such as the code of a spike protein. The Vaxzevria (previously COVID-19 vaccine AstraZeneca/Oxford) is an adenovirus vaccine (ChAdOx) that has been authorized by the EMA since January 2021. It is a chimpanzee adenovirus-vectored vaccine encoding the SARS CoV 2 spike glycoprotein (ChAdOx1-S, ChAdOx1 nCoV-19), manufactured by the Serum Institute of India and SKBio [158]. Results from four clinical trials in the United Kingdom, Brazil, and South Africa showed that Vaxzevria was safe and effective at preventing COVID-19, as well as resulting in robust neutralizing antibody and T-cell responses [159,160,161,162]. In March 2021, with the increase in vaccination across the population, some serious adverse reactions occurred with ChAdOx1 nCov-19 (AstraZeneca) and Janssen (Johnson & Johnson) vaccines. In very rare cases, their use has led to the development of immune thrombotic thrombocytopenia (very similar to heparin-induced autoimmune thrombocytopenia, HIT) caused by anti-platelet factor 4 (anti-PF4) antibodies that activate platelets. These pathological changes may cause unusual clotting such as cerebral venous thrombosis [163,164]. More data on this pathophysiology are therefore crucial for preventing these harmful effects. Nonetheless, EMA issued a statement that the benefits of Vaxzevria continue to outweigh its risks, and the vaccine can continue to be administered.

The COVID-19 vaccine Janssen is another vector vaccine, developed by Johnson & Johnson, that received authorization in the EU in March 2021 [165]. This vaccine is composed of replication-incompetent human adenovirus that encodes a SARS-CoV-2 full-length spike glycoprotein and provokes a similar immune response after vaccination as Vaxzevria.

For the Sputnik V vaccine, which is already registered in more than 55 countries, an EMA rolling review started on the 4th March [166,167]. Unlike the COVID-19 vaccine Janssen, Sputnik V includes two different types of human adenovirus vectors (rAd26 and rAd5), which ensure lasting immunity [168].

All approved vaccines within the EU are safe, while differing in efficacy ranging from 72 to 95%. Most vaccines are given in two doses, except for Johnson & Johnson (Janssen), who state that a single dose will provide protection against the disease. Likewise, they differ in other factors such as time required for full immune response, protection against COVID-19 in the aged and young population, suitability for the elderly, and storage properties (Appendix A).

Expectations from EU-approved vaccines to produce nationwide immunity against COVID-19 as well as identifying their side effects and possible health risks need to be monitored in further large-scale randomized clinical trials.

Several vaccines have been developed and tested in other countries apart from in Europe. India‘s first COVD-19 vaccine Covaxin (Bharat Biotech) is an inactivated vaccine developed using whole-virion-inactivated, Vero cell-derived platform technology. It has demonstrated 81% efficacy [169]. These types of inactivated vaccines have also been developed in China, namely, Synopharm and Sinovac, showing similar efficacy in COVID-19 prevention [170]. American Novavax has developed a unique protein-based vaccine, NVX-CoV2373 (trade name COVOVAX), requiring two doses with an efficacy of 86% (UK variant) and 60% (South African variant) [171].

Vaccination is crucial for achieving a sufficient level of protection against the virus, especially for immunocompromised patients and patients with comorbidities. For this reason, cancer patients who are at an even higher risk of severe COVID-19 infection should be prioritized for vaccination against SARS-CoV-2. These patients are advised to use mRNA vaccines (BNT162b2, Pfizer-BioNTech vaccine), which have a better safety profile and therefore a lower risk of adverse reactions in these patients [172]. However, the efficacy of these human vaccines in these patients is questionable [173]. The BNT162b2 vaccine was shown to be effective and safe in a study of 134 patients and older adults with various frailty and disability profiles, providing protection regardless of their condition [174]. On the other hand, several studies have found lower levels of antibodies in patients with multiple myeloma after the first dose of this vaccine than in the vaccine trials [175,176]. These findings further increase the emphasis of the second dose in cancer patients. However, large prospective and well-designed clinical trials regarding efficacy and safety among immunocompromised patients are necessary.

## 5. Conclusions

COVID-19, a global pandemic, is causing substantial health and economic damage. With continuous emergence of new mutations, which are proving to be more infectious, it is becoming increasingly clear that isolation and quarantine measures on their own are not sufficient to contain virus spread. More than a year after the onset of the pandemic, there is a great need for standardized treatment management, effective prevention, and use of first-line drugs to conduct therapeutic interventions during severe cases. The EMA is gradually assessing and registering more vaccines, which differ in terms of mechanisms of action, number of doses, or method of storage. As of 18 February 2021, at least 86 vaccine candidates are being tested in phase III clinical trials, and around seven vaccines have been approved by several countries (https://covid19.trackvaccines.org/ accessed on 18 February 2021). Some drugs, such as antimalarial medication (hydroxychloroquine), used at the beginning of the COVID-19 pandemic were blocked in clinical trials due to serious side effects. Recently, however, the EMA authority advised against the use of ivermectin for the prevention and treatment of COVID-19 patients. On the other hand, due to the good results of the studies, the use of glucocorticoids is recommended for patients with severe symptoms who are on oxygen therapy or pulmonary ventilation. Dexamethasone may suppress cytokine storm and reduce inflammation, preventing mortality in the most affected group of patients. Immunotherapy, either through convalescent plasma transplantation or monoclonal antibodies, could provide an important bridge between vaccines and the treatments that have already been recommended. In addition, for innovative approaches including peptides, ACE-2 inhibitors, and cell therapy, large and randomized clinical trials are still necessary. The most critical tools to intervene in the COVID-19 pandemic are safe and effective vaccines, available for all age groups, with minimal side effects. Therefore, acquiring deep knowledge of the mechanism of action of therapeutics against this virus is essential.

## Figures and Tables

**Figure 1 pathogens-10-00636-f001:**
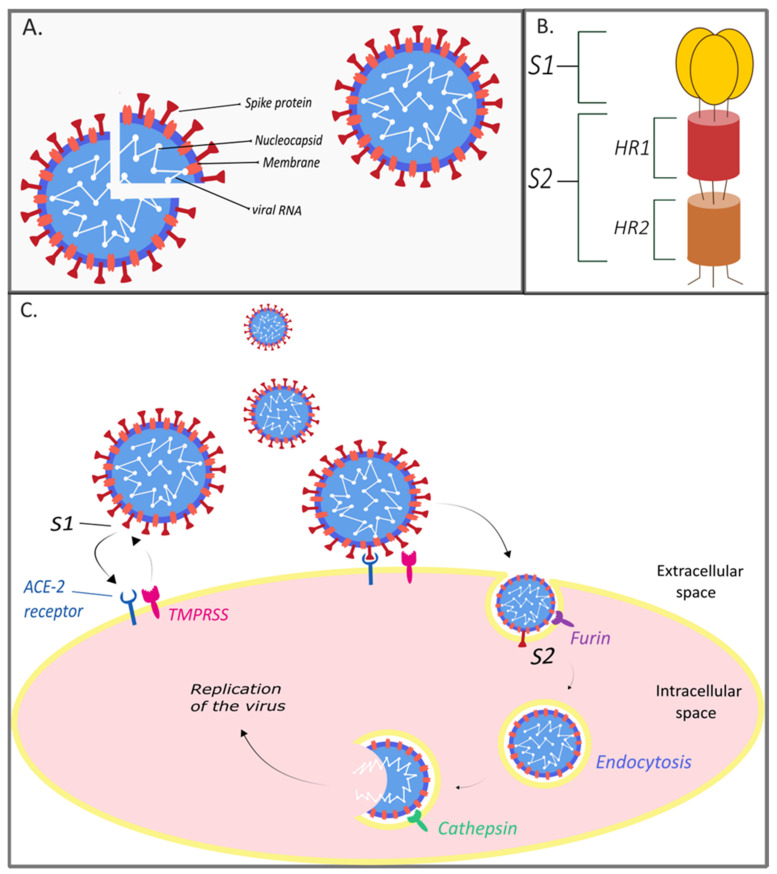
Schematic structure of SARS-CoV-2 and schematic of spike protein and depiction of the SARS-CoV-2 viral entry. (**A**) Schematic structure of SARS-CoV-2. The viral surface proteins including spike protein (S) and membrane protein (M); these proteins are ingrained in a lipid envelope. The single-stranded viral RNA is linked to the nucleocapsid protein. (**B**) Schematic of spike protein. Schematic structure of viral spike protein showing subunits S1 and S2 and domains HR1 and HR2. (**C**) Schematic of SARS-CoV-2 viral entry. S protein targets the host ACE-2 (angiotensin-converting enzyme 2) receptor and enters in the host cell. Note the binding between viral subunits S1/S2 and ACE-2 receptor, as well as cleavage by cell surface protein TMPRSS2 protease. Host cell proteases furin and cathepsin also take part in the cleavage. For the viral particle to enter into the host cell (endocytosis), S protein cleaving into S1 and S2 subunits at S1/S2 cleavage site is essential. This happens either by the serine protease TMPRSS2, or by endosomal proteases Cathepsin B/L [38]. Spike protein can also be cleaved by furin convertases [39]. After the fusion with the host cell membrane, viral genome is released, translated, and replicated. Upon protein assembly, exocytosis takes place, which releases viral particles from the cell. Authors of this article created this figure using Inkscape software; it is not based on any previously published image.

**Figure 2 pathogens-10-00636-f002:**
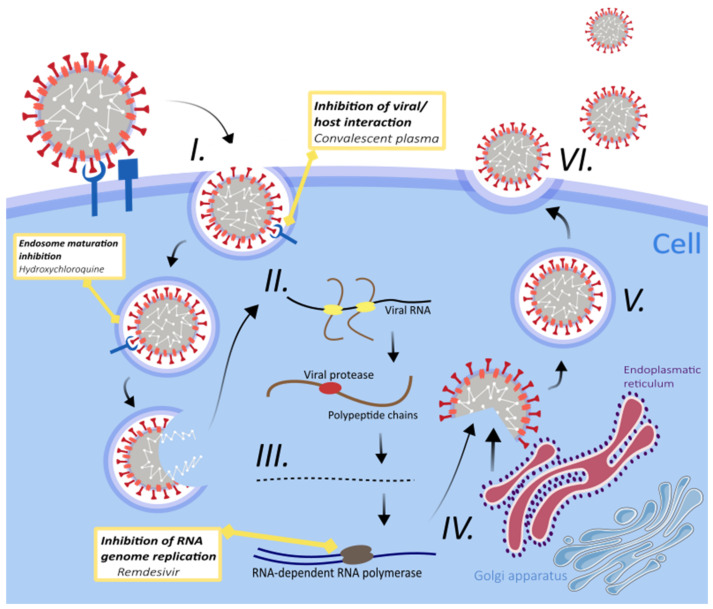
The lifecycle of SARS-CoV-2 in infected host cells. (**I.**) Viral entry. The virus enters through direct interaction between the viral spike protein (S protein) and the cellular ACE-2 (angiotensin-converting enzyme 2) receptor. (**II.**) Translation. Afterwards, the viral genome is released and translated into viral replicase polyproteins (PP1a, PP1ab). (**III.**) Viral protease cleavage. The polyproteins are then cleaved by viral proteases into functional proteins [97]. (**IV**.) Translation and replication of RNA. The replication of the viral genome is facilitated by viral replication complex, including RNA-dependent RNA polymerase. The process of transcription and replication occurs in complex membranes adjacent to double-membrane vesicles (DMVs), derived from the rough endoplasmic reticulum. (**V**.) Packaging. Viral nucleocapsids are composed of packaged viral genomes with positive sense RNA and translated viral structural proteins. (**VI**.) Release of a virion. Enveloped virion is exported from the cell by exocytosis. Potential mechanisms of action with targets for antiviral interventions are blocking the interaction of virus and host cell via the use of antibodies (convalescent plasma therapy), inhibiting the maturation of endosomes through the use of hydroxychloroquine, and use of nucleoside or nucleotide analogues including remdesivir in inhibiting viral genome replication of viral genome. Authors of this article created this figure using Inkscape software; it is not based on any previously published image.

## Data Availability

No new data were created or analyzed in this study. Data sharing is not applicable to this article.

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
