# Peer review of "Fundamental and Advanced Therapies, Vaccine Development against SARS-CoV-2"

_pathogens, 2021, doi:10.3390/pathogens10060636_

Round 1

Reviewer 1 Report

A reasonably well written review. A few grammatical errors in introduction. Additional inclusions suggested as below would strengthen the manuscript better.

  1. Include evidence from Remdesivir RCTs – ACTT-1 and Solidarity
  2. Include evidence from studies on convalescent plasma by Joyner M et al, PLACID trial
  3. Include details of other vaccines - Covaxin (Bharat biontech), Novavax, Sinopharm, Sinovac etc.

Author Response

Response to Reviewer 1

We thank the reviewer for valuable advices, comments and suggestions that will improve the quality of our manuscript.

We have taken all comments into account and adjusted the text in line with these comments.

1. Include evidence from Remdesivir RCTs – ACTT-1 and Solidarity

Response:  We have included paragraph (yellow highlighted) in the section 3.1.2. Remdesivir., describing two large Remdesivir RCTs – ACTT-1 and Solidarity studies,

Pages 5-6, lines-195-203

2. Include evidence from studies on convalescent plasma by Joyner M et al, PLACID trial

Response:  We have included paragraph (yellow highlighted) in the section 3.1.5. Passive immunity therapy describing both recommended studies

Page 7, lines 292-301

3. Include details of other vaccines - Covaxin (Bharat biontech), Novavax, Sinopharm, Sinovac etc.

Response:  We have included paragraph (yellow highlighted) in the section 4.1.  Vaccine development- describing suggested vaccines

Page 14, lines 607-614

Reviewer 2 Report

In this interesting review the authors summarize the gradual development of various treatments and vaccines, as they emerged, a year after the outbreak of the pandemic. A few points should be taken into consideration:

  • Please revise the whole manuscript for minor grammar and syntax errors
  • Antimalarial drugs: Please comment the fact that these drugs were universally accepted as an anti-COVID-19 treatment at the early stages of the pandemic before the results of randomized clinical trials that proved no benefit.
  • Remdesivir: Please provide more details on the clinical data from the clinical trials supporting its benefit on patients with COVID-19
  • Convalescent plasma: Please also include data from the systematic review and meta-analysis by Janiaud et al. PMID: 33635310
  • Please also include a section on supportive fundamental therapies including thromboprophylaxis, bronchodilators/vasodilators, mechanical ventilation/high nasal flow/ECMO. (See also relevant sections in PMID: 33128197)
  • Please elaborate on the indications of use of monoclonal antibodies some of which have gained accelerated approval such as bamlanivimab and etesevimab.
  • Please elaborate on the putative pathophysiology regarding thrombosis following vaccination with the AZ and JJ vaccines
  • Please refer to "Supplemental Table 1" and not "Table 1" since it will be included as a supplemental file
  • Please discuss the anticipated vaccine efficacy in view of the emerging SARS-CoV-2 variants, as well as efficacy and safety in special populations such as patients with cancer (See pmids: PMID: 33623886, PMID: 33861315, PMID: 33887255, PMID: 33812495, PMID: 33768521)

Author Response

Response to Reviewer 2

We thank the reviewer for valuable comments and suggestions in order to improve the quality of our manuscript.

We have taken all comments into account and adjusted the text in line with the comments.

  • Please revise the whole manuscript for minor grammar and syntax errors

Response:  The manuscript was revised for English grammar, please see the revised text.

  • Antimalarial drugs: Please comment the fact that these drugs were universally accepted as an anti-COVID-19 treatment at the early stages of the pandemic before the results of randomized clinical trials that proved no benefit.

Response:  we have added corresponding comment in the section 3.1.1. Antimalarial drugs, documenting their authorization on March 2020 by FDA, followed by their withdrawal due to adverse effects

Page 5, Lines 163-165

  • Remdesivir: Please provide more details on the clinical data from the clinical trials supporting its benefit on patients with COVID-19

Response:  we have included paragraph (yellow highlighted) in the section 3.1.2. Remdesivir, describing Remdesivir RCTs – ACTT-1 and Solidarity clinical trials, revealing different clinical data, due to totally different design of both studies

Pages 5-6, lines-195-203

  • Convalescent plasma: Please also include data from the systematic review and meta-analysis by Janiaud et al. PMID: 33635310

Response:  Please find included paragraph (yellow highlighted) in the section 3.1.5. Passive immunity therapy describing data from suggested review Janiaud et al. PMID: 33635310

Page 7, lines 292-301

  • Please also include a section on supportive fundamental therapies including thromboprophylaxis, bronchodilators/vasodilators, mechanical ventilation/high nasal flow/ECMO. (See also relevant sections in PMID: 33128197)

Response:  We have added a new section  3.1.6. Supportive therapies, mentioning all suggested therapies important by for reducing the concomitant risks of COVID-19

Pages 9-10, lines 357-387

  • Please elaborate on the indications of use of monoclonal antibodies some of which have gained accelerated approval such as bamlanivimab and etesevimab.

Response:  Please find response to approval of suggested monoclonal antibodies

Pages 7-8, lines 306-310

  • Please elaborate on the putative pathophysiology regarding thrombosis following vaccination with the AZ and JJ vaccines

Response:  We have included paragraph describing the health risks based on thrombosis pathology in rare patients vaccinated with the AstraZeneca and Jassen (Johnson & Johnson) vaccines

Page 14, lines 579-588

  • Please refer to "Supplemental Table 1" and not "Table 1" since it will be included as a supplemental file

Response:  We have corrected in the text.

Pages:  14, line 603

            15, line  649

  • Please discuss the anticipated vaccine efficacy in view of the emerging SARS-CoV-2 variants, as well as efficacy and safety in special populations such as patients with cancer (See pmids: PMID: 33623886, PMID: 33861315, PMID: 33887255, PMID: 33812495, PMID: 33768521)

Response:  Please find short paragraph dedicated to this topic.

Page 15, lines 615-628